# MoCA-Video: Motion-Aware Concept Alignment for Consistent Video Editing

## Abstract

We present MoCA-Video, a training-free framework for semantic mixing in videos. Operating in the latent space of a frozen video diffusion model, MoCA-Video utilizes class-agnostic segmentation with diagonal denoising scheduler to localize and track the target object across frames. To ensure temporal stability under semantic shifts, we introduce momentum-based correction to approximate novel hybrid distributions beyond trained data distribution, alongside a light gamma residual module that smooths out visual artifacts. We evaluate model's performance using SSIM, LPIPS, and a proposed metric, CASS, which quantifies semantic alignment between reference and output. Extensive evaluation demonstrates that our model consistently outperforms both training-free and trained baselines, achieving superior semantic mixing and temporal coherence without retraining. Results establish that structured manipulation of diffusion noise trajectories enables controllable and high-quality video editing under semantic shifts.

## 1 Introduction

Diffusion models Ho et al. (2020); Rombach et al. (2021) have revolutionized image synthesis and enabled controllable video generation. Video Diffusion Models Ho et al. (2022) introduced coherent frame synthesis, while subsequent works Singer et al. (2022); Chen et al. (2024); Wan et al. (2025); Kong et al. (2024); Chen et al. (2023) enhanced visual quality and temporal coherence. These advances have spawned diverse applications including image-to-video animation Xing et al. (2023); Guo et al. (2024), subject-driven editing Ku et al. (2024), and affordance insertion Kulal et al. (2023).

However, current video generation approaches remain fundamentally constrained by existing training data distributions, limiting their ability to create novel hybrid entities that combine characteristics from multiple semantic categories. This limitation becomes particularly evident in video *semantic mixing*, which is the task of composing hybrid visual entities in video by selectively blending semantic concepts from multiple source inputs, e.g., creating a "cat-astronaut" by fusing feline features with space suit elements. The goal is to generate coherent and consistent hybrid objects that preserve key structural properties while adopting complementary semantics across temporal domains.

While semantic mixing has been explored in static images through MagicMix Liew et al. (2022) and FreeBlend Zhou et al. (2025), extending this capability to videos presents unique challenges. Existing video editing methods rely primarily on frame-by-frame operations or global style transfers, failing to achieve fine-grained, region-specific semantic mixing. Prompt-based and auxiliary-based strategies, in particular, often blur the distinction between local and global features, making it difficult to isolate their effects. To tackle the challenge, we introduce MoCA-Video (Motion-Aware Concept Alignment in Video), a training-free framework that addresses this challenge through structured manipulation of latent noise trajectories. Given an input video and reference image, it injects reference image semantic features into the video, producing temporally consistent hybrid entities that transcend the limitations of existing diffusion model training distributions. (see Section 3)

We compare MoCA-Video against current baselines both quantitatively and qualitatively. Quantitatively, we employ SSIMWang et al. (2004), LPIPS(I/T)Zhang et al. (2018) and a newly introduced metric CASS (Conceptual Alignment Shift Score) based on CLIPRadford et al. (2021) , and its normalized variant that compensates for inherent task difficulty biases. Qualitatively, we demonstrate visual results on dataset derived from FreeBlend and extended by DAVIS entities, showing more visually compelling and temporally coherent semantic mixes than prior methods (see Section 4)

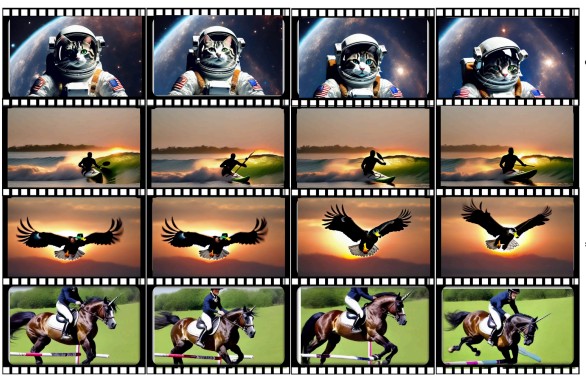

Figure 1: MoCA-Video enables diverse semantic mixing across object categories. Given a reference image and an input video along with global and local prompts, the method outputs semantically mixed videos that blend concepts from both the image and video inputs, e.g., surfer with kayak.

Our contributions are summarized as follows:

**MoCA-Video Framework.**   We introduce the first training-free framework for video semantic mixing via latent noise manipulation. The framework adopts IoU-based object tracking, momentum-corrected denoising approximation and gamma residual stabilization for semantic mixing.

**Task-specific metrics.**   We propose CASS and relCASS, CLIP-based metrics for semantic mixing evaluation, providing robust assessment across intra-class and inter-class blending scenarios.

**Experimental validation.**   Extensive benchmarking on training-free (FreeBlend, RAVE) and pre-trained methods (AnimateDiffV2V, TokenFlow (PnP, SDEdit)) indicates that MoCA-Video achieves superior performance across visual fidelity, temporal coherence, and semantic alignment.

## 2   RELATED WORK

### 2.1   SEMANTIC MIXING AND VIDEO CONCEPT COMBINATION

Semantic mixing, first introduced in the image domain through MagicMix and later improved by FreeBlend, focusing on blending disparate concepts into coherent, novel objects. These methods exploit the denoising dynamics of diffusion models to factorize layout and content or apply staged latent interpolation for more stable blending. However, both approaches are limited to static images and overlook temporal consistency. MagicEdit extends semantic editing into video by injecting prompt-driven features at specific diffusion stages. While it maintains motion to some degree, it lacks spatial control, and explicit fusion of image-based semantics. Our work fills these gaps with a training-free framework MoCA-Video that combines image conditioning, latent-space mask tracking and motion correction to deliver semantical blendings that remains coherent across time.

### 2.2   IDENTITY-PRESERVED VIDEO GENERATION

One similar track of creative video generation is identity-preserved text-to-video generation (IPT2V), focusing on retaining a reference subject's appearance while generating new motion. ID-Animator He et al. (2024) enables a zero-shot face-driven videos but often overfits to the input image. ConsisIDYuan et al. (2024) and EchoVideoWei et al. (2025) further enhances face detail but limited to human identities. In contrast to IPT2V, video semantic mixing focus on combine the reference subject into existing video subject rather than preserving a single identity at specific region.

### 2.3   VIDEO INPAINTING

Video inpainting methods extend region specific filling to the temporal domain, typically relying on optical flow, feature correspondences, or domain priors to maintain frame-to-frame consistency. DIVEHuang et al. (2024) uses DINO features and LoRA-based identity registration for subject-driven edits, and ObjectMateWinter et al. (2024) builds an Object Recurrence Prior to train on large supervised composition. TokenFlowGeyer et al. (2023) propagates self-attention tokens via nearest neighbor matching to enforce smooth transitions, while RAVEKara et al. (2023) shuffles

Figure 2: MoCA-Video pipeline. Given a base video (astronaut) and reference image (cat), we recover the latent trajectory via DDIM inversion. At selected timesteps, we segment the target object with GroundedSAM2 using RGB proxy of predicted clean image, and track masks via IoU-maximization. Reference features are injected into masked regions, followed by momentum correction to approximate the denoising of manipulated data distribution and gamma noise stabilization.

latent and condition grids during denoising to maintain consistency upon unshuffling. Compare to video inpainting, MoCA-Video operates on region specific edits; however, instead of replacing the whole segmented area, it preserves the original features and blends them with new semantic content.

## 3 METHODOLOGY

MoCA-Video enables semantic mixing in videos by seamlessly blending features from a reference image into a target object within base video, while preserving global scene layout and motion consistency from the original video. Built upon a frozen text-to-video diffusion model VideoCrafter2, which is initialized from Stable Diffusion 2.1, our approach introduce a structured video editing pipeline that manipulates the latent noise trajectory rather than performing frame-by-frame edits.

Given a generated video and a reference image, the method first recovers the base video's latent trajectory via DDIM inversion. At chosen steps when the target object has emerged but remains semantically flexible, we employ Grounded-SAM2 to estimate soft masks on predicted clean frame ($x_0$), localizing the target object using an IoU-based maximization algorithm 1. These masks define a "fusion zone", within which reference features are injected into the latent representation. To maintain temporal coherence, MoCA-Video adopts the diagonal denoising scheduler of FIFODiffusion Kim et al. (2024), enabling consecutive frame updates to share semantic information effectively.

Beyond this backbone, two lightweight mechanisms are used to enhance and stabilize the blending process: (i) momentum correction, which approximates denoising trajectories perturbed by semantic shifts; and (ii) a gamma residual noise module that smooths out flicker and local artifacts by injecting calibrated low-scale residual noise; making MoCA-Video excels in quality hybrid entity appearance.

### 3.1 LATENT SPACE TRACKING

At the core of MoCA-Video lies the ability to blend object semantics directly within the latent space of the diffusion model. To enable localized feature injection, we first identify and track the target object across the video latents. Let $\mathbf{X}$ denote the sequence of noisy latent representations obtained via DDIM inversion. For target object identification, we decode the predicted clean image $x_0$ from the latent at chosen timesteps using it as a proxy RGB frame for segmentation. Despite residual noise, $x_0$ preserves sufficient semantic structure, particularly at later timesteps, enabling reliable object detection. We apply a class-agnostic segmentation model (Grounded-SAM2) to this decoded frame, yielding a binary mask $m_0$ in pixel space. This mask is then mapped back into latent space serving as an auxiliary condition to define the subregion $x_m$ that corresponds to the target object.

To propagate the masked region across temporal frames, we adopt an IoU-based tracking strategy using overlap maximization Danelljan et al. (2019) (See Alg. 1). This design is essential for two critical reasons: (1) segmentation operating on noisy intermediate representations, where object boundaries

are ambiguous, would impose additional difficulties; (2) as denoising advances and edited objects become visually sharper, segmentation becomes increasingly challenging under ongoing semantic mixing procedures. Maintaining consistent mask propagation therefore crucial to prevent spatial drift and ensure manipulated fusion regions remain stably tracked across the entire denoising steps.

For each timestep $t$, the current segmentation mask $m_t$ is predicted and compared with the previous mask $m_{t-1}$. If the IoU exceeds a predefined threshold $\tau$, we updates the mask with the new prediction; otherwise, we retain the previous mask. This produces a sequence of masks $\mathbf{X}^m = \{x_0^m, x_1^m, \ldots, x_{t'-1}^m\}$ that stably track the target region from timestep $t'$ to the final step.

The resulting mask sequence serves as an auxiliary spatial gate that guides the denoising process, restricting feature injection to the target regions while maintaining the integrity of the surrounding representation. At timestep $t$, this gating is realized by combining the base latent $x_t$ with the conditioned latent $x_t^{\text{cond}}$ encoded from the reference image through the autoencoder:

$$x_t^{\text{mix}} = x_t \cdot (1 - x_t^m) + \lambda \cdot x_t^{\text{cond}} \cdot x_t^m$$

Here, $x_t^{\text{mix}}$ represents the fused latent, where the mask $x_t^m$ defines a soft fusion zone rather than strict boundaries. This design enables MoCA-Video to tolerate segmentation imperfections, as feature mixing occurs within denoising process, where the DDIM scheduler inherently smooths out minor mask errors and outperforms precise pixel-space replacement, which often introduces visible artifacts when masks are imprecise. Importantly, feature injection intensity ($\lambda = \frac{t}{1000}$) is not uniform across timesteps. Peak injection happens around $t'$, when the object has emerged but remains semantically editable, then gradually decreases as denoising progresses so that the original video features will not be overwritten by reference image. This adaptive weighting ensures that major semantic blending occurs during the optimal window automatically. Algorithm 1 implements this tracking process.

### 3.2 Adaptive Motion Correction with Momentum

While latent tracking ensures consistent spatial localization of the target object, it does not guarantee that the blended appearance evolves smoothly across time. Without additional constraints, feature injection will cause abrupt changes or motion-induced artifacts that break temporal coherence and visual fidelity. To tackle this, we introduce a momentum-corrected DDIM denoising algorithm 2 that approximates the denoising trajectory under semantic shifts due to limited training data distribution.

**Momentum-Corrected DDIM.** Recall that the standard DDIM update Song et al. (2022) predicted clean image $\hat{x}_0^{(\text{DDIM})}$ at timestep $t$ and updates the latent $x_{t-1}$ using a directional term $\text{dir}_t$ derived from the noise estimate:

$$x_{t-1} = \sqrt{\alpha_{t-1}} \underbrace{\left( \frac{x_t - \sqrt{1-\alpha_t}\, \epsilon_\theta^{(t)}(x_t)}{\sqrt{\alpha_t}} \right)}_{\hat{x}_0^{(\text{DDIM})}} + \underbrace{\sqrt{1 - \alpha_{t-1} - \sigma_t^2}\, \epsilon_\theta^{(t)}(x_t)}_{\text{dir}_t} + \sigma_t\, \epsilon_t.$$

In MoCA-Video, we augment this process with a momentum term $v_t$ that accumulates residual changes across timesteps to stabilize the denoising trajectories perturbed by semantic injection:

$$\hat{x}_0^{(\text{corr})} = \hat{x}_0^{(\text{DDIM})} + \kappa_t v_t, \qquad v_t = \beta v_{t-1} + (1 - \beta) g_t$$

, where $g_t = x_t - x_{t-1} + \lambda \text{dir}_t$ models the deviation introduced by semantic feature injection, $\beta$ controls momentum decay, and $\kappa_t$ gradually decreases with $t$ to prevent over-correction at later denoising stages when fine details are being refined. The term $x_t - x_{t-1}$ functions as a geometric correction to the standard DDIM directional vector $\text{dir}_t$. As semantic injection brings positional difference, leading to a new directional component that deviates from the original denoising trajectory. When combined with $\lambda$, $\text{dir}_t$, the resulting update $g_t$ points toward a novel trajectory that approximates the hybrid distribution enabling the generation of semantically blended entities such as an astronaut with cat features or a corgi-shaped coffee machine. It actively reorients the denoising process toward data distributions that lie outside the training manifold of the base diffusion model. This geometrically-guided heuristic enables MoCA-Video to navigate toward previously unseen semantic combinations while maintaining the structural coherence of the underlying diffusion dynamics.

**Algorithm 1** Tracking by Overlap Maximization

**Require:** Sequence of latents $\{x_0, x_1, \ldots, x_t\}$
**Require:** Initial Object mask $m_0 \leftarrow SEG(x_0)$
**Require:** Set value for $\tau$
1: **for** $t = 1$ to $t' - 1$ **do**
2:     $m \leftarrow SEG(x_t)$
3:     $iou \leftarrow \text{IoU}(m, m_{t-1})$
4:     **if** $iou > \tau$ **then**
5:        $m_t \leftarrow m$
6:     **else**
7:        $m_t \leftarrow m_{t-1}$ {Retain previous mask to avoid drift}
8:     **end if**
9: **end for**

**Algorithm 2** Momentum-Corrected Denoising

**Require:** $\{x_0, x_1, \ldots, x_t\}, \text{dir}_t, \beta, \lambda, \kappa_0, T$
1: Initialize $v_t \leftarrow 0$
2: **for** $t = T, \ldots, 1$ **do**
3:     $\hat{x}_0^{(\text{DDIM})} \leftarrow \dfrac{x_t - \sqrt{1 - \alpha_t}\, \epsilon_\theta^{(t)}(x_t)}{\sqrt{\alpha_t}}$
4:     $x_{t-1}^{(\text{DDIM})} \leftarrow \sqrt{\alpha_{t-1}}\, \hat{x}_0^{(\text{DDIM})} + \text{dir}_t + \sigma_t\, \epsilon_t$
5:     $g_t \leftarrow x_t - x_{t-1}^{(DDIM)} + \lambda\, \text{dir}_t$
6:     $v_t \leftarrow \beta v_{t-1} + (1 - \beta)g_t$
7:     $\hat{x}_0^{(\text{corr})} \leftarrow \hat{x}_0^{(\text{DDIM})} + \kappa_0\big(1 - \tfrac{t}{T}\big)v_t$
8:     $x_{t-1} \leftarrow \sqrt{\alpha_{t-1}}\, \hat{x}_0^{(\text{corr})} + \text{dir}_t + \sigma_t\, \epsilon_t$
9: **end for**
10: **Return** $\{x_{t-1}\}_{t=1}^{T}$

**Gamma Residual Noise.** To further stabilize the denoising trajectory, we inject a small $\gamma$-scaled noise term at each step:

$$x_t^{\text{final}} = x_t^{\text{mix}} + \gamma \cdot \epsilon, \qquad \epsilon \sim \mathcal{N}(0, I),$$

where $\gamma$ controls the residual strength. This gamma residual mechanism serves as a lightweight regularizer that damps unstable fluctuations introduced by semantic injection and momentum correction, mitigating inter-frame flicker artifacts. In conjunction with momentum correction, the regularization ensures that semantic blending transformation evolves smoothly across temporal sequences.

## 4 EXPERIMENTS

To the best of our knowledge, MoCA-Video is the first framework that systematically addresses the problem of *video entity blending*. Given the absence of existing benchmark for this task, we construct an evaluation dataset tailored for assessing entity-level semantic blending performance.

### 4.1 ENTITY BLENDING DATASET

Our dataset builds upon the broad super-categories introduced in the CTIB dataset Zhou et al. (2025), *i.e.* Transports, Animals, Common Objects, and Nature, which have been validated as comprehensive coverage of the most salient real-world concepts. To further enhance object diversity, we incorporate annotated classes from the DAVIS-16 video segmentation dataset Perazzi et al. (2016). This integration proves particularly valuable as DAVIS-16 was specifically curated to minimize semantic overlap between annotated objects, ensuring that target entities cannot be trivially identified through class labels alone and requiring more sophisticated semantic understanding for successful blending. We organize the DAVIS-16 into 16 additional subcategories under super-category. Using this taxonomy, we design evaluation pairs spanning both intra-category combinations (e.g., cow and sheep) and inter-category pairs with substantial semantic distance (e.g., astronaut and cat). This systematic approach as shown in Fig. 3 yields 100 unique entity pairs for comprehensive benchmarking of our proposed framework's performance.

Tab 1 shows that each dataset entry consists of: (1) a source prompt; (2) a base video generated from the source prompt; (3) a reference image; (4) a scalar blending strength that controls the intensity of feature injection from reference into the video. When reference images are unavailable, we generate it using Stable Diffusion 2.1. The blending strength, $\lambda$, determines the degree of feature transfer, where higher values impose stronger feature injection from the reference image and vice versa.

### 4.2 EVALUATION

We propose evaluating *video entity blending* approaches along three complementary axes: (i) structural consistency, (ii) temporal coherence, and (iii) semantic integration quality.



Figure 3: Source and conditioned objects are paired through prompt generation, base video creation, and conditioned image creation. We design intra-category and inter-category blends using FreeBlend super-categories and DAVIS-16 sub-categories. The pipeline is extensible, allowing new super- and sub-categories for custom datasets.

Table 1: **Dataset Examples:** Source Prompts, Targeted Objects, Conditioned Prompt, and Conditioning Strength.

| Source Prompt | Object | Conditioned Prompt | $\lambda$ |
|---|---|---|---|
| A cute teddy bear with soft brown fur and a red bow tie | Teddy | the condition is a hamster | 2.0 |
| A blooming rose garden in full color under morning sunlight | Rose | the condition is lavender | 1.5 |
| A colorful tropical fish swimming in a coral reef | Fish | the condition is a dolphin | 1.2 |

For fidelity and smoothness, we adopt widely used perceptual metrics. SSIM and LPIPS-I to measure frame-level similarity between the generated and base videos. LPIPS-T computes perceptual differences between adjacent frames to quantify temporal smoothness and stability.

While these metrics effectively capture appearance quality and motion consistency, they do not directly assess the quality of semantic blending operations. To this end, we propose the CASS (Conceptual Alignment Shift Score) metric, a novel CLIP-based metric for quantifying semantic integration in video entity blending. CASS measures how the semantic alignment of a video shifts before and after blending, relative to both the original text prompt and the conditioned image.

Formally, let $V_{\text{orig}}$ denote the original video, $V_{\text{fused}}$ the fused video. and $I_{\text{cond}}$ the reference image. We denote by $E(\cdot)$ the CLIP visual encoder and by $L(\cdot)$ the CLIP text encoder. We compute:

$$\text{CLIP-T}_{\text{orig}} = \text{sim}(E(V_{\text{orig}}), L_{\text{orig}}) \qquad \text{CLIP-I}_{\text{orig}} = \text{sim}(E(V_{\text{orig}}), E(I_{\text{cond}}))$$
$$\text{CLIP-T}_{\text{fused}} = \text{sim}(E(V_{\text{fused}}), L_{\text{orig}}) \qquad \text{CLIP-I}_{\text{fused}} = \text{sim}(E(V_{\text{fused}}), E(I_{\text{cond}}))$$

In a quality semantic mixing outcome: the original video aligns strongly with the text prompt, but weakly with the reference image. After blending, these roles reverse, CLIP-T decreases as the model moves away from the prompt, while CLIP-I increases as features from the reference image are integrated. Hence, we design CASS to capture this complementary shift:

$$\text{CASS} = (\text{CLIP-I}_{\text{fused}} - \text{CLIP-I}_{\text{orig}}) - (\text{CLIP-T}_{\text{fused}} - \text{CLIP-T}_{\text{orig}})$$

For varying difficulties, we further compute the relative relCASS. As intra-category blends represent easier cases where the base video already shares semantic similarity with the reference image, while inter-category blends are more challenging since the baseline similarity is low. By normalizing each shift relative to its original alignment score, relCASS provides a non-biased evaluation of framework performance regardless of the underlying blend difficulties. A higher CASS and relCASS values signify better semantic mixing toward the reference image while remains original features.

$$\text{rel\_CLIP-I} = \frac{\text{CLIP-I}_{\text{fused}} - \text{CLIP-I}_{\text{orig}}}{\text{CLIP-I}_{\text{orig}}}, \quad \text{rel\_CLIP-T} = \frac{\text{CLIP-T}_{\text{fused}} - \text{CLIP-T}_{\text{orig}}}{\text{CLIP-T}_{\text{orig}}}$$
$$\text{relCASS} = \text{rel\_CLIP-I} - \text{rel\_CLIP-T}.$$

### 4.3 BASELINE COMPARISONS

We compare MoCA-Video against both training-free and pretrained video diffusion models.

**Training-free methods** We evaluate (i) FreeBlend, originally designed for image semantic mixing, which we adapt by applying frame edits and subsequently animated into video sequences for fair comparison; and (ii) RAVE, which directly extends reference-image guidance to video editing.

Table 2: **Quantitative comparison**. Training-free methods capture injected semantics but overwhelm original content. Pretrained methods preserve structure and motion but suppress semantic injection. MoCA-Video achieves best balance across visual, temporal and task-specific score

| Method | SSIM ↑ | LPIPS-I ↑ | LPIPS-T ↓ | CASS ↑ | relCASS ↑ |
|---|---|---|---|---|---|
| Pretrained | | | | | |
|   AnimateDiffV2V | 0.74 | 0.19 | **0.01** | 0.68 | 0.57 |
|   TokenFlow PnP | **0.93** | 0.02 | **0.01** | 2.87 | 0.07 |
|   TokenFlow SDEdit | 0.27 | **0.82** | 0.15 | 1.98 | 0.02 |
| Training-free | | | | | |
|   FreeBlend + Dynamicrafter | 0.34 | 0.62 | 0.16 | 1.47 | 0.37 |
|   RAVE | 0.61 | 0.37 | 0.04 | 3.80 | 0.13 |
|   AnyV2V | 0.69 | 0.17 | 0.02 | 2.31 | 0.42 |
| **MoCA-Video (ours)** | 0.35 | 0.67 | 0.11 | **4.93** | **1.23** |

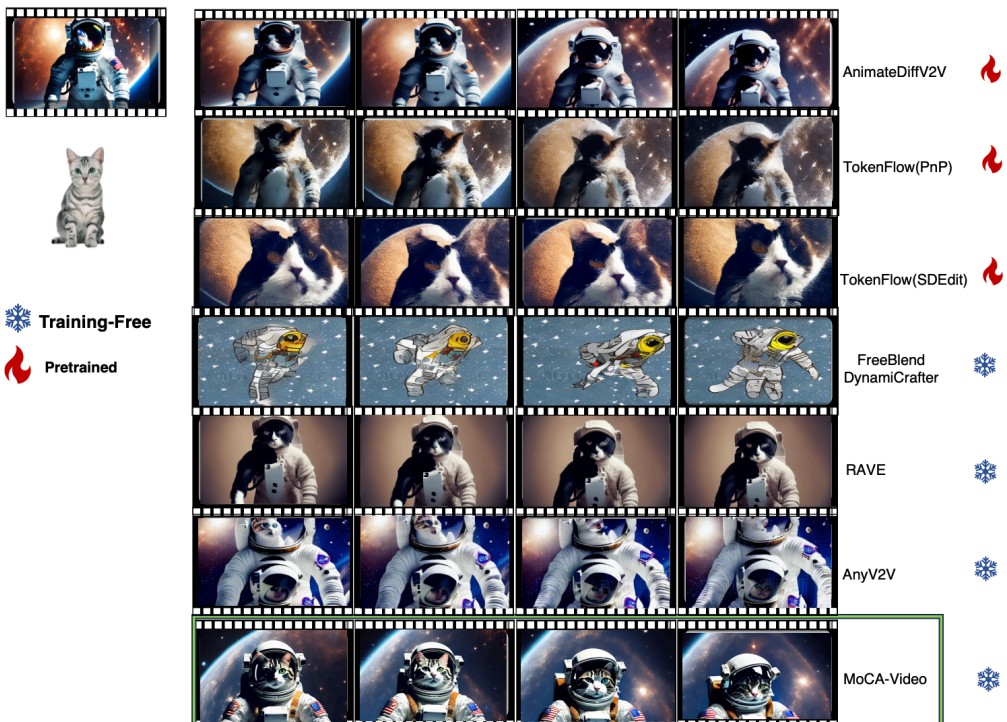

Figure 4: Visual comparison MoCA-Video achieves coherent fusion with stable semantics and smooth motion. While others either harm visual fidelity or present weak semantic mixing.

**Pretrained methods** We include (i) AnimateDiffV2V, which generates edited sequences conditioned on the source prompt and prioritize smooth motion dynamics; and (ii) TokenFlow under two configurations: PnP through self-attention map consistency and SDEdit via partial denoising.

Table 2 reveals that pretrained methods excel at preserving spatial fidelity and motion smoothness but consistently fail to achieve meaningful semantic integration. AnimateDiffV2V presents the highest SSIM (0.74) and smoothest temporal transition (LPIPS-T = 0.01), but virtually no semantic transformation (CASS = 0.68). TokenFlow PnP enforces structure preservation with negligible semantic transformation and SDEdit introduces visual artifacts yielding a lower semantic alignment score.

Training-free methods demonstrates stronger edits but with significant trade-off. FreeBlend extended by DynamiCrafter shows moderate semantic mixing (CASS = 1.47) accompanied by substantial temporal inconsistency, while RAVE achieves stronger semantic transfer (CASS = 3.80) but with substantial loss of the fine-grained detail information from the original object's features.

Figure 5: **Visual ablation study.** Visualization shows that without IoU tracking, object drift; without motion correction, causes frame misalignment; and without gamma noise brings edge flicker.

MoCA-Video achieves the most effective semantic blending performance (CASS = 4.93 improve averagely *rel.* $\Delta$ =56% , relCASS = 1.23, with average of *rel.* $\Delta$ =81% improvement, which indicates MoCA-Video perform better across multiple difficulty level of prompts) while maintaining competitive perceptual quality (SSIM = 0.35 decreased by *rel.* $\Delta$ =-35%, but LPIPS-I = 0.67 increased by *rel.* $\Delta$ =40%) and minimal temporal artifacts (LPIPS-T = 0.11 *rel.* $\Delta$ =-32%). These results demonstrate MoCA-Video's unique capability for semantic integration, with minimal and tolerable trade-off on structural fidelity, and temporal coherence, establishing a comprehensive benchmark across both training-free and pretrained methodologies for video semantic mixing.

## 4.4 ABLATION STUDIES

We conduct comprehensive ablation studies to validate the necessity of each component in MoCA-Video across three critical dimensions: (i) core architectural modules, (ii) robustness to mask quality, and (iii) generalization beyond curated prompts.

### 4.4.1 CORE MODULE ANALYSIS

We systematically ablate three key components: overlap maximization for mask tracking, momentum motion correction, and gamma residual stabilization. Table 3 reveals that IoU-based overlap maximization contributes most significantly to performance, with its removal causing substantial degradation in spatial fidelity (SSIM: *rel.* $\Delta$ =-20%) and semantic alignment (CASS: *rel.* $\Delta$ =-33%). Motion correction proves critical for temporal stability, with its absence increasing jitter substantially (LPIPS-T: *rel.* $\Delta$ =-39%). By reorienting denoising trajectories toward hybrid distributions introduced by semantic injection, this mechanism stabilizes temporal evolution and ensures blended objects maintain coherence across frames. Finally, removing gamma residual noise leads to edge flickering and fine-detail instability despite preserving global structure. Figure 5 provides visual confirmation that each component addresses distinct failure modes in semantic video mixing.

Table 3: Ablation results for MoCA-Video components. IoU-based overlap maximization has the largest impact on spatial fidelity and semantic alignment. Motion correction reduces temporal jitter and misalignment. Gamma residual noise smooths out edge flicker and boundary dimness.

| Method Variant | SSIM ↑ | LPIPS-I ↑ | LPIPS-T ↓ | CASS ↑ | relCASS ↑ |
|---|---|---|---|---|---|
| Full MoCA-Video | 0.35 | 0.67 | 0.11 | 4.93 | 1.23 |
| w/o Overlap Maximization (IoU) | **0.28** | **0.63** | **0.20** | **2.90** | **0.75** |
| w/o Motion Correction | 0.30 | 0.65 | 0.18 | 3.10 | 0.80 |
| w/o Gamma Residual Noise | 0.32 | 0.66 | 0.15 | 4.20 | 1.10 |

Table 4: Mask robustness. Imperfect mask region doesn't harm significantly on semantic mixing.

| Method | SSIM ↑ | LPIPS-I ↑ | LPIPS-T ↓ | CASS ↑ | relCASS ↑ |
|---|---|---|---|---|---|
| GroundDINO (BBox, lower boundary) | 0.52 | 0.69 | 0.16 | 2.69 | 0.13 |
| GroundedSAM2 (Ours) | 0.35 | 0.67 | 0.11 | 4.93 | 1.23 |

### 4.4.2 ROBUSTNESS TO IMPERFECT SEGMENTATION

Since MoCA-Video relies on inference-time segmentation masks, we evaluate robustness under suboptimal conditions. Table 4 compares performance using region specific GroundedSAM2 masks against coarse bounding boxes from GroundingDINO. Even with imprecise supervision, MoCA-Video maintains superior semantic mixing performance compared to all baselines, demonstrating that latent-space diffusion manipulation exhibits inherent tolerance to segmentation imperfections.

### 4.4.3 GENERALIZATION BEYOND CURATED DATA

To assess scalability of the curated dataset, we extend evaluation beyond the curated dataset by incorporating multi-object scenes and additional object categories following our dataset construction pipeline. Table 5 shows that performance degrades modestly on complex scenes (CASS: *rel.* $\Delta =$-24%), yet semantic mixing remains robust even with multiple distracting objects. This validates that our framework generalizes effectively, enabling researchers to construct custom datasets using the same taxonomic approach while maintaining consistent performance across diverse scenarios.

These ablation results collectively demonstrate that MoCA-Video's design choices are well-motivated and functioned, with each component targeted at specific challenges in video semantic mixing while maintaining robustness across varying conditions and scene complexities.

Table 5: Prompt robustness. Multiple objects in the video do not significantly harm performance, achieving comparable scores to the baseline model, proving the extensibility of the prompt dataset.

| Setting | SSIM ↑ | LPIPS-I ↑ | LPIPS-T ↓ | CASS ↑ | relCASS ↑ |
|---|---|---|---|---|---|
| Multi-object prompts | 0.45 | 0.65 | 0.12 | 3.74 | 0.08 |
| Original dataset | 0.35 | 0.67 | 0.11 | 4.93 | 1.23 |

## 5 CONCLUSION

We presented MoCA-Video, the first training-free framework for video semantic mixing. Operating through structured manipulation of latent noise trajectories, our method integrates (1) IoU-based overlap maximization for consistent object tracking; (2) momentum-corrected denoising for approximating novel hybrid distributions; and (3) gamma residual noise stabilization for fine-grained temporal smoothness. Extensive experiments show that MoCA-Video outperforms both training-free and pretrained methods, achieving stronger semantic blending without compromising motion or visual quality, while demonstrating tolerance to imperfect masking and dataset extensibility. MoCA-Video establishes structured noise-space manipulation as a promising paradigm for controllable video synthesis that transcends the limitations of existing diffusion model training distributions.

**Reproducibility Statement** We have taken several steps to ensure the reproducibility of our work. The experimental setup, including model architectures, training hyperparameter, and dataset pre-processing, is described in detail in Sections 3 and 4. Additional implementation details, and hyper-parameter settings are provided in the Appendix **??**. To further facilitate reproducibility, we include a link to the source code and instructions for running the experiments in Appendix **??**. Together, these resources are intended to make it straightforward for researchers to replicate our results and build upon our method.

**Ethics Statement** This work does not involve human subjects, personally identifiable information, or sensitive data. All datasets used are publicly available and employed in accordance with their respective licenses. The proposed methodology is intended solely for academic research and poses no foreseeable risks of misuse, harmful applications, or ethical concerns beyond standard con-

siderations in machine learning research. We have adhered to the ICLR Code of Ethics throughout the preparation and submission of this work.

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
