# 6 APPENDIX

## 6.1 ADDITIONAL VISUAL RESULTS

We provide additional qualitative comparisons on concept-blending tasks, Bird+Cat, Surfer+Kayak, Horse+Unicorn, and Cow+Sheep, each driven by the same global and conditioned prompts and input reference video and image. Across all examples, MoCA-Video achieves the best trade-off between semantic fusion, temporal smoothness, and frame-level consistency. AnimateDiffV2V largely preserves the original video with only minimal blending of the new concept, while Free-Blend+DynamiCrafter can merge the two concepts but suffers from temporal jitter, spatial artifacts, and a lack of overall visual quality. These results underscore MoCA-Video's strong ability to inject a merge novel semantics into video while maintaining both motion coherence and aesthetic quality.

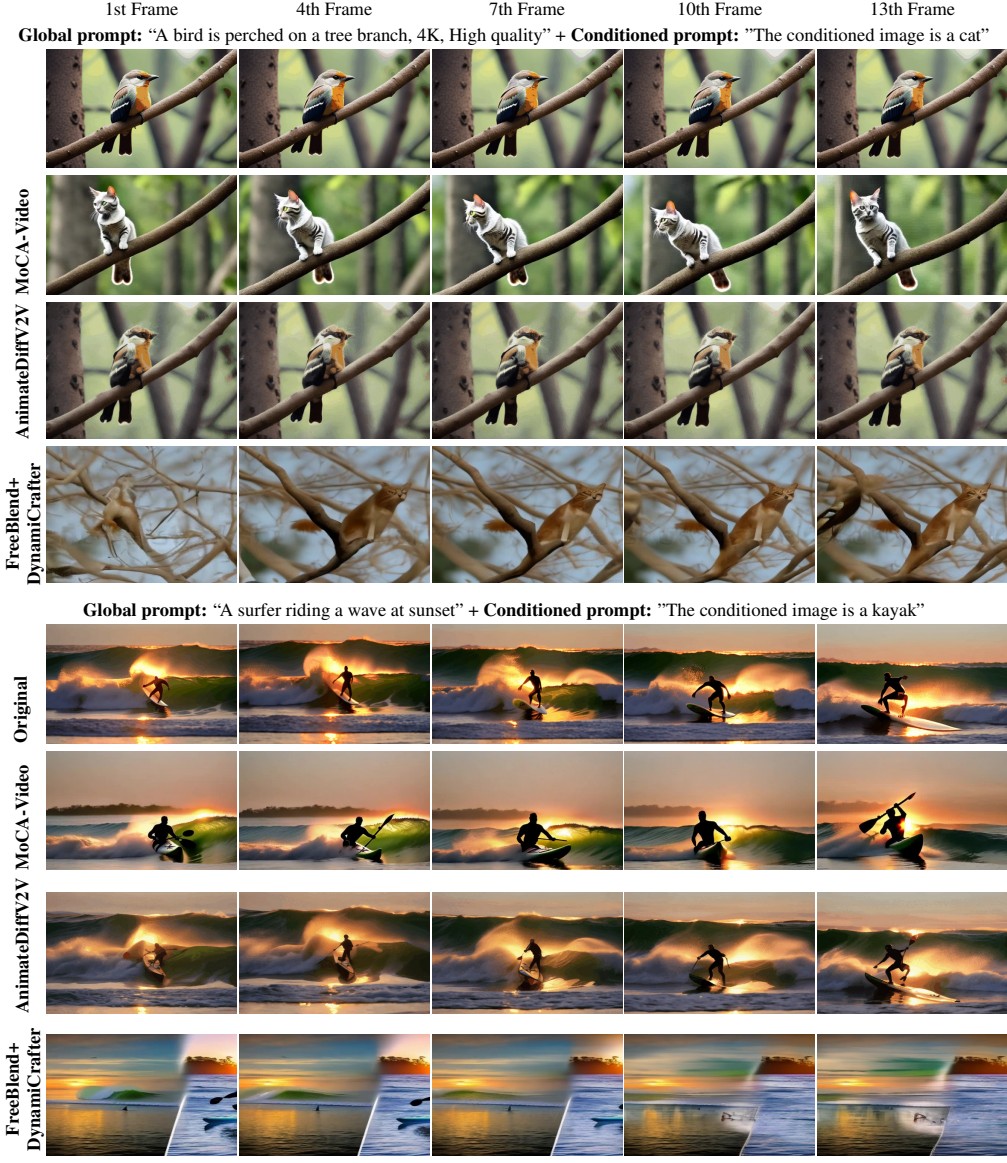

Figure 6: **Multi-sample Qualitative Comparison.** We show four different prompts (two blocks above, two more below in the full paper) across five evenly-spaced frames, comparing Original, MoCA-Video, AnimateDiffV2V, and FreeBlend+DynamiCrafter in each block.

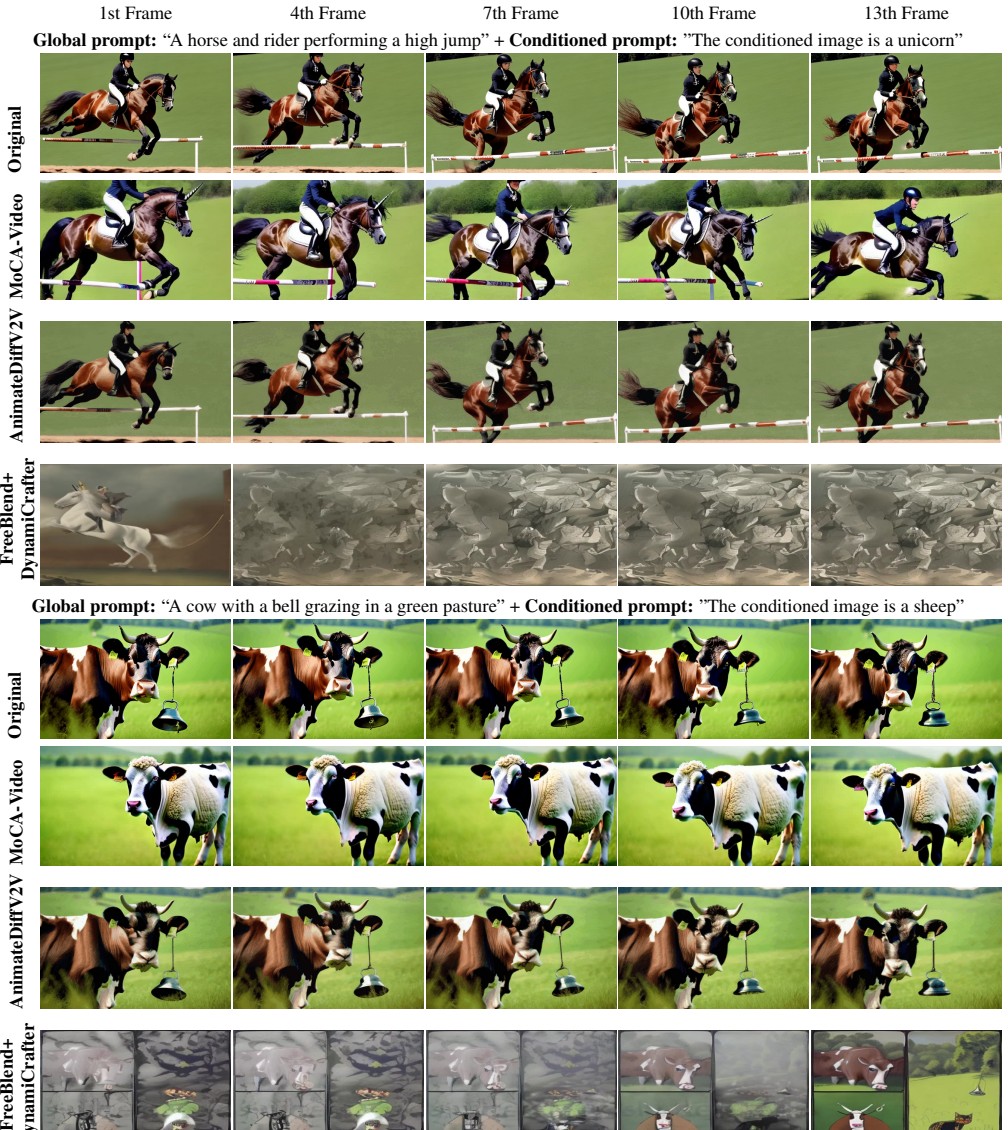

Figure 7: **Multi-sample Qualitative Comparison.** Continued example from Figure 6.

## 6.2 FREEBLEND CLIP-METRIC ANALYSIS

Here we analyze FreeBlend's CLIP score (based on absolute difference) and compare with our CASS proposal. The main drawback with FreeBlend's "CLIP-BS" metric is that it treats the blended video's similarity to each of the two original prompts in isolation, and then it simply takes the absolute difference of it. In practice, CLIP-BS rewards cases where the fused clip simply "looks like" one of the two prompts (*i.e* high mix_score) or even shows both concepts side by side, rather than effectively blending them.:

$$\text{mix\_score} = \text{sim}(V_{\text{fused}}, \text{"a photo of A"}) + \text{sim}(V_{\text{fused}}, \text{"a photo of B"}),$$
$$\text{original\_score} = \text{sim}(V_A, \text{"a photo of A"}) + \text{sim}(V_B, \text{"a photo of B"}),$$
$$\text{CLIP\_BS} = \big|\text{mix\_score} - \text{original\_score}\big|.$$

The original video's CLIP score is already high (*i.e.*, the generated video matches its own prompt), therefore a high CLIP-BS can be obtained in two ways: (1) The mixed score exceeds the original

score, (2) The mixed score becomes very low (near zero), when calculating the difference of mixed score and original score will present a large negative value that will be positive after applying the absolute value.

Clearly, the second scenario is problematic as an image with no similarity to any of the mixed concepts will have a high CLIP-BS score, trivially reflecting the alignment of the original video to the the prompt used to create it.

The first scenario also allows for sub-optimal visual mixtures with high score CLIP-BS. When $\text{sim}(V_{fused}, \text{"A photo of A"}) \geq \text{original\_score}$, we can obtain a high CLIP-BS score, fully ignoring the presence of concept B in the generated video. Likewise, if $\text{sim}(V_{fused}, \text{"A photo of B"}) \geq \text{original\_score}$ would score high in CLIP-BS fully disregarding the alignment with concept A. This also means that a "fused" video where concept A and B are depicted as independent objects (**i.e.** without any meaningful combination) can also be favored with a high CLIP-BS score.

Scenario 1 can yield high CLIP-BS scores even when there is no effective mixing of the concept, the mere presence of both objects is largely favor in CLIP-BS. Also very low scores for $\text{sim}(V_{fused}, \text{"A photo of A"})$ could be offset by large scores in $\text{sim}(V_{fused}, \text{"A photo of B"})$ and vice-versa. This is undesirable for the blending task,

Table 6: Comparison of semantic mixing methods FreeBlend metrics.

| Method | CLIP-BS↑ | DINO-BS↑ |
|---|---|---|
| FreeBlend | 6.65 | 0.27 |
| MoCA-Video | 4.00 | 0.12 |

We evaluated both metrics on the same 100 test samples. Although FreeBlend's score appears higher, its visual results are noticeably inferior to ours, highlighting the shortcomings of their absolute difference measure. By contrast, our CASS metric explicitly captures the desired semantic shift: we compare the original video's alignment to its own prompt before and after fusion (which drops as new content is injected) and we separately track its alignment to the conditioned image (which rises after blending). As a result, CASS only produces high values when the fused video truly moves away from the source concept and toward the new concept, faithfully reflecting genuine, high-quality semantic mixing.

### 6.3 HUMAN EVALUATION PROTOCOL

#### 6.3.1 USER STUDY

We recruited twenty volunteers (aged 18–45, balanced gender) from our university community, all of whom reported normal or corrected-to-normal vision and no prior involvement in this project. Each participant completed eight independent trials in a single session lasting approximately twenty minutes. At the start, participants provided electronic consent and reviewed a brief demonstration trial to familiarize themselves with the interface and rating criteria. In each trial, participants first viewed an input video along with an input image that shows two different concept. Immediately after, three anonymized 2-second clips are shown, each generated by one of the methods (MoCA-Video or baselines), presented in random order. Participants rated each clip on four dimensions using a 1-5 Likert scale, where 1 is the worst scale while 5 is the best scale: Blending Quality (how well the clip fused the input video and input image), Video Consistency (smoothness and temporal coherence of motion), Character Consistency (stable blended character consistency), and Overall Quality (general visual fidelity and appeal). We presented these as a multiple-choice grid so that every video received a score for each criterion.

Figure 8 shows that the human evaluation complements our automated metrics by capturing subjective judgments of semantic mixing quality, motion coherence, character consistency and overall attractiveness and creativeness of semantic concept blending, critical factors in assessing the real-world effectiveness of video concept-blending methods.

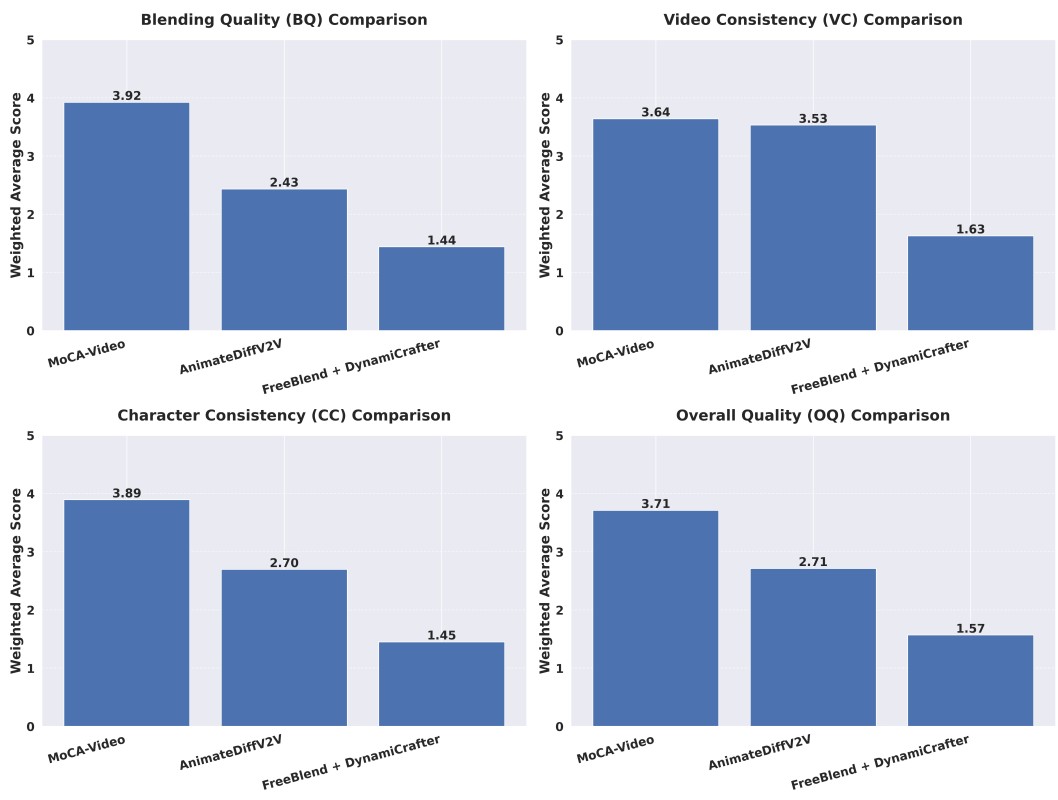

Figure 8: The plot highlights that MoCA-Video outperforms the other methods on Blending Quality and Overall Quality, while still delivering strong Video Consistency and Character Consistency. AnimateDiffV2V scores highest on both consistency measures—reflecting its conservative editing—but lags on blending and overall appeal. FreeBlend+DynamiCrafter ranks lowest across all four metrics, confirming it struggles to balance concept fusion with temporal and character fidelity.

## 6.4 EXPERIMENTAL SETUP

For our empirical maximization, we ran all experiments on a single NVIDIA V100 GPU (32 GB RAM) with Python 3.10, PyTorch 2.1, and CUDA 11.8. Each 147-frame video requires roughly 45 minutes of inference (batch size = 1) and peaks at about 10 GB of VRAM. To ensure the semantic injection has fully taken effect, we sample frames from the midpoint of the generated sequence, which is the around $\left\lfloor \dfrac{\mathtt{new\_video\_length}}{2} \right\rfloor$ so that our evaluations reflect the final, fully blended result. We tuned the following key hyper-parameters to balance semantic fusion and temporal fidelity: an injection timestep of $t' = 300$, conditioning strength $\gamma \in [1.5, 2.0]$, IoU threshold = 0.5, momentum decay $\beta = 0.9$, and base correction weight $\kappa_0 \in [1.0, 2.0]$. Other diffusion-scheduler and denoising-step settings were left at their defaults.

### 6.4.1 COMPUTATIONAL EFFICIENCY ANALYSIS

We report per-frame inference time for all compared methods in Table 7 (excluding preprocessing such as model initialization, base video generation, and feature extraction). Our method requires 17s per frame, which decomposes into 13s for FIFO-Diffusion's diagonal denoising scheduler and 4s for MoCA-specific operations (IoU tracking and mask extraction from $t'$ to $t = 0$).

The computational overhead primarily stems from the diagonal denoising scheduler (13s), which is essential for maintaining temporal coherence during semantic injection. The additional MoCA-specific operations add only 4s per frame, representing a modest 31% overhead relative to the base FIFO-Diffusion framework. This overhead is necessary and well-justified: the diagonal denoising design enables forward-referencing across frames that is critical for achieving smooth semantic

Table 7: Per-frame inference time comparison (in seconds).*6s for staged feedback-driven image blending + 4s for video animation.

| Method | Time (s/frame) |
|---|---|
| AnimateDiff | 1 |
| TokenFlow (PnP/SDEdit) | 6 |
| FreeBlend + DynamiCrafter* | 10 |
| RAVE | 3 |
| AnyV2V | 5 |
| **MoCA-Video (ours)** | **17** |
| - FIFO-Diffusion diagonal denoising | 13 |
| - MoCA components (IoU + segmentation) | 4 |

mixing with motion preservation, which are capabilities that faster methods fundamentally lack, as evidenced by their significantly lower CASS scores (Table 2). While methods like AnimateDiff (1s) and RAVE (3s) are faster, they achieve CASS scores of only 0.68 and 3.80 respectively, compared to our 4.93, demonstrating that our approach achieves superior semantic mixing quality at reasonable computational cost.

### 6.5 THEORETICAL JUSTIFICATION FOR MOMENTUM CORRECTION

While our paper primarily demonstrates that structured manipulation of diffusion noise trajectories can achieve controllable and high-quality semantic mixed video, we provide theoretical grounding for Alg. 2 momentum correction mechanism, which shows substantial empirical improvements (39% reduction in LPIPS-T, Table 3).

**Problem Setup.** Let $p_{\text{base}}(x_0)$ denote the data distribution of the original video and $p_{\text{ref}}(x_0)$ the distribution of reference image features. Our goal is to sample from a hybrid distribution $p_{\text{hybrid}}(x_0)$ that blends characteristics from both distributions in a spatially-localized manner defined by mask $m$.

**Hybrid Latent Construction.** Standard DDIM approximates $\mathbb{E}[x_0|x_t]$ under $p_{\text{base}}$. Our masked injection creates:

$$x_t^{\text{mix}} = x_t \odot (1 - m) + \lambda x_t^{\text{cond}} \odot m, \tag{1}$$

shifting $x_t$ toward a manifold where the masked region aligns with $p_{\text{ref}}$. Here, $\lambda = \frac{t}{1000}$ is time-variant, emphasizing feature injection at earlier timesteps (when fine details are absent) while diminishing at later steps to avoid overwriting.

**Trajectory Deviation Capture.** The correction term $g_t = x_t - x_{t-1} + \lambda \cdot \text{dir}_t$ captures the deviation between:

- $x_{t-1}$ in Alg. 2 line 4: the standard DDIM prediction under $p_{\text{base}}$
- $x_{t-1}$ in Alg. 2 line 8: the desired prediction under $p_{\text{hybrid}}$

Thus, $g_t \approx \Delta x_t^{\text{hybrid}} - \Delta x_t^{\text{base}}$, representing the instantaneous directional shift induced by semantic injection.

**Connection to Score Functions.** Under reverse diffusion, the instantaneous velocity is governed by:

$$\frac{dx_t}{dt} \propto \nabla_{x_t} \log p(x_t). \tag{2}$$

For discrete DDIM steps with $\Delta \alpha = \alpha_{t-1} - \alpha_t > 0$, substituting Equation (12) from the DDIM paper Song et al. (2022) yields:

$$x_{t-1} - x_t \approx \Delta \alpha \cdot [\text{drift terms involving } \nabla_{x_t} \log p(x_t)]. \tag{3}$$

Therefore:

$$g_t \approx \Delta \alpha \left[ \nabla_{x_t} \log p_{\text{hybrid}}(x_t) - \nabla_{x_t} \log p_{\text{base}}(x_t) \right], \tag{4}$$

which is the *instantaneous score drift* at timestep $t$.

**Momentum as Score Drift Accumulation.** By maintaining $v_t = \beta v_{t-1} + (1 - \beta)g_t$, we compute an exponentially-weighted estimate of the persistent directional bias. Expanding recursively:

$$v_t = (1 - \beta) \sum_{k=0}^{T-t} \beta^k g_{t-k}. \tag{5}$$

For high momentum ($\beta \approx 1$) during initial injection phases, this approximates temporal smoothing:

$$v_t \approx \frac{1}{T - t + 1} \sum_{\tau=0}^{t} g_\tau \approx \mathbb{E}_{\tau \leq t}[g_\tau]. \tag{6}$$

Substituting Equation (4):

$$v_t \approx \mathbb{E}_{\tau \leq t} \left[ \nabla_{x_\tau} \log p_{\text{hybrid}}(x_\tau) - \nabla_{x_\tau} \log p_{\text{base}}(x_\tau) \right], \tag{7}$$

the *accumulated score drift* along the trajectory.

**Corrected Prediction.** The update $\hat{x}_0^{(\text{corr})} = \hat{x}_0^{(\text{DDIM})} + \kappa_t v_t$ adjusts the clean image prediction toward the mode of $p_{\text{hybrid}}$, where $\kappa_t = \frac{t}{1000}$ provides time-dependent weighting that emphasizes corrections early in denoising (when semantic structure is established) and diminishes them later (when fine details are refined).

Overall, Alg. 2 implements a *first-order approximation to score-matching under distribution shift*, where momentum $v_t$ estimates the cumulative score deviation caused by feature injection. While not an exact sampler for $p_{\text{hybrid}}$ (which would require training data), it provides a principled heuristic that geometrically interpolates between known distributions, validated by our empirical results (Table 3).

### 6.6 DIAGONAL DENOISING SCHEDULER IMPLEMENTATION

To clarify how the FIFO-Diffusion Kim et al. (2024) scheduler is implemented in our video editing process, we provide an explicit explanation of how diagonal denoising achieves temporally coherent semantic mixing.

**Queue Structure.** The diagonal denoising scheduler processes video frames in a queue:

$$Q = \{z_1^{\tau_1}, z_2^{\tau_2}, \ldots, z_{nf}^{\tau_{nf}}\}, \tag{8}$$

where each frame $i$ is at a different noise level $\tau_i$ with $0 < \tau_1 < \tau_2 < \ldots < \tau_{nf} = T$. Unlike standard parallel denoising where all frames share the same timestep, this diagonal arrangement enables frames to reference cleaner (lower noise) neighbors during denoising.

**Partitioned Denoising.** At each iteration, the queue is partitioned into $n$ blocks of size $f$ (the base model's temporal capacity window):

$$Q = [Q_0, Q_1, \ldots, Q_{n-1}], \tag{9}$$

where each block $Q_k$ is denoised via:

$$Q_k \leftarrow \Phi(Q_k, \tau_k, c; \epsilon_\theta), \tag{10}$$

with $\tau_k = \{\tau_{kf+1}, \ldots, \tau_{(k+1)f}\}$ and $\Phi(\cdot)$ denoting the DDIM sampler.

**Semantic Injection.** During denoising, we inject reference features through masked blending:

$$z_i^{\text{mix}} = z_i^{\text{video}} \odot (1 - m_i) + \lambda \cdot z_{\text{ref}} \odot m_i, \tag{11}$$

where $m_i$ is the tracked mask from Alg. 1, $z_{\text{ref}}$ is the encoded reference image, and $\lambda$ controls injection strength.

**FIFO Queue Management.** After each iteration:

1. **Dequeue**: The fully denoised frame $z_1^{\tau_0}$ at the queue head is removed and output

2. **Enqueue**: A new noisy latent $z_{i+nf}^{\tau_{nf}}$, initialized using the dequeued frame, is added to the tail

3. **Propagation**: All remaining frames shift forward, creating a sliding temporal window

**Temporal Coherence Mechanism.** This diagonal pattern enables temporal propagation of semantic edits: as frame $i$ moves through the queue from high noise ($\tau_{nf}$) to clean ($\tau_0$), it continuously observes already-blended neighboring frames. This ensures smooth temporal transitions with stable identity alignment across video sequences. Crucially, each frame references both *cleaner* preceding frames (providing semantic guidance) and *noisier* following frames (maintaining motion context) during denoising.

**Advantage over Standard Denoising.** Compared to traditional window-based denoisers (e.g., Open-Sora), where all latents in a temporal window are denoised simultaneously at the same timestep $t$, our diagonal scheduler processes frames at different noise levels $\{\tau_1, \ldots, \tau_{nf}\}$. This heterogeneous noise structure makes temporal relationships more robust under identity-changing scenarios, as cleaner frames provide stable reference points for noisier ones undergoing semantic transformation.

## 6.7 CODE RELEASE

Anonymous code can be found here: `https://anonymous.4open.science/r/MoCA-Video-5DD7/`, we recommend accessing it with Google Chrome for the best user experience. We will publish the official code soon after acceptance.

## 6.8 THE USE OF LARGE LANGUAGE MODELS (LLMS).

We used commercial LLM (ChatGPT, Claude) for editorial polishing and generating dataset sample prompts. The AI assisted with improving writing clarity and academic style, and created diverse example prompts for our evaluation dataset. All technical contributions, experimental results, and scientific conclusions are entirely our own work. AI-generated content was manually reviewed and validated by the authors.