# OpenReview forum: "Motion-Aware Concept Alignment for Consistent Video Editing"
_ICLR.cc/2026/Conference — Submitted to ICLR 2026_

### Official Review · Reviewer_hc8D · 2025-11-02

**Soundness:** 2
**Presentation:** 2
**Contribution:** 1
**Rating:** 4
**Confidence:** 4

**Summary:**

This paper tackles video semantic mixing, which is blending a reference image’s concept into a target object in a base video, without retraining the diffusion model. The proposed MoCA-Video pipeline uses (i) DDIM inversion of a frozen text-to-video model (VideoCrafter2), (ii) class-agnostic segmentation + IoU-based overlap maximization to track the edited region in latent space, (iii) latent feature injection from the reference image within a “soft” mask, (iv) a diagonal (FIFO-style) denoising scheduler to propagate edits temporally, (v) a momentum-corrected DDIM update to stabilize trajectories perturbed by semantic shifts, and (vi) a lightweight gamma residual noise regularizer to reduce flicker.

**Strengths:**

- **Training-free approach and simple design**
The paper provides a controllable hybrid concept mixing without fine-tuning the whole model, operating purely at inference time, which reduces the resource consumption compared with training-based methods. Combining soft masked latent fusion, overlap-max tracking, a diagonal scheduler, and a momentum term is conceptually straightforward.

- **Wide ablations**
The paper tests imperfect masks and multi-object scenes.

**Weaknesses:**

- **Incremental novelty**
The core ingredients of this paper, which are DDIM inversion plus masked latent injection and consistency propagation, are similar to image mixing (MagicMix, FreeBlend) and video editing propagation (TokenFlow, AnyV2V, and other recent tuning-free V2V pipelines). The paper claims to be the first training-free video semantic mixing, but the difference from tuning-free region-aware video editing is unclear and questionable. A sharper novelty boundary and deeper comparison to the most relevant video editing methods are needed.

- **Efficiency and practicality issues**
The pipeline requires DDIM inversion per video, and long sequences (147 frames) take 45 minutes on a V100. That is heavy for training-free editing. A runtime comparison vs. baselines is needed.

- **Missing details**
How the reference image latent is computed and aligned (spatial rescaling, aspect mismatch) for injection at time t? How robust is tracking under large appearance changes? How the diagonal FIFO scheduler is implemented in the editing process?

The problem is interesting and practical, and the pipeline is coherent and training-free. However, the contribution is incremental and overclaimed; core designs are common in previous works. I lean toward a negative score.

**Questions:**

See weaknesses

---

> ### Author Response · Authors · 2025-11-25
> **Clarification on Novelty and Core Contributions**
>
> We thank the reviewer for the feedback and for raising these concerns. We would like to provide further clarification and alleviate the issues you highlighted.
>
> Regarding the novelty concern, we would like to emphasize that first, while they share certain implementation, video semantic mixing is not video editing. Editing method either preserve identity or replace content. While video semantic mixing is requires dual-feature preservation, which is in lack of all the compared baseline video editing approach (both trained and training-free approaches). That’s why traditional metric is not enough to evaluate the task, and we therefore propose CASS and relCASS to evaluate this new task. Secondly, contribution wise, DDIM inversion and mask conditions are very standard way in video editing tasks, but it’s way not enough to achieve the task of semantic mixing, that’s why we say in the beginning that they share certain implementation, but it’s not the same. Beyond the shared video editing implementation, the main contribution of MoCA-Video is to close the gap between traditional video editing and this novel video semantic mixing tasks. To do that, we propose a new pipeline, with IoU latent tracking, momentum correction and gamma residual stabilization to maintain a temporally stabled mixed identity video generation. As reviewer mention the previous paper FreeBlend and MagicMix, these two are in image domain, but not video domain, which is in lack of temporal stableness handling, and it’s not trivial given the ablation study in **Table 3**, without it causing a 37% CASS loss. And from the visual result we can also observe the motion drift and fail to combine the two given concept.
>
> We also have complete baseline comparison with both quantitative and qualitative result presented in the paper, including training-free approaches FreeBlend and RAVE; and pretrained approaches: AnimateDiffV2V, TokenFlow (PnP & SDEdit). Quantitatively, we show that our result achieves the best result among all the comparison in **Table 2**, and from the baseline **Figure 4**, we show the difference between ours and others, ours give the most appealing visual appearance compared to either failed new feature injection or complete replacement.
>
> We additionally evaluated **AnyV2V** as reviewer suggested., an image-edit-then-propagate method whose performance is fundamentally constrained by the quality of the initial frame edit. Since AnyV2V relies on an external image editing pipeline before video propagation, it achieves good structural preservation (LPIPS-I: 0.17, LPIPS-T: 0.02) but struggles with semantic mixing due to limitations in the image editing stage. The scores for AnyV2V are: SSIM: 0.69, LPIPS-I: 0.17, LPIPS-T: 0.02, CASS: 2.31, relCASS: 0.42. Notably, its CASS score of 2.31 is **53%** lower than our approach (CASS: 4.93), confirming that frame-by-frame editing followed by propagation cannot achieve the dual-feature preservation required for true semantic mixing. We have updated the result in both **Table 2** and **Figure 4** for quantitative and qualitative comparison.
>
> Hence, in conclusion, as Reviewer **8fGT** and Reviewer **WcLF** both mentioned in the strength of our paper, video semantic mixing is a novel defined task, which is an under-explored problem and non-trivial, which needs to solve the temporal coherence under dual-concept fusion. The effectiveness of the approach is also validated by our rich baseline comparison, both quantitatively and qualitatively.

---

> ### Author Response · Authors · 2025-11-25
> **Methodology Clarification and Detailed Example**
>
> To help readers to understand the paper easier, we would like to explain further the details of our approach and give example to showcase it.
>
> As Figure 2 our main pipeline figure shows, the reference image is encoded through the same VAE encoder used by the video diffusion model. More explicitly to define it, the input reference image is $I_{ref} \in \mathbb{R}^{H\times W \times 3}$, which is an arbitrary resolution in RGB. In order to align the size of reference image with the video frame resolution, we resize the $I_{ref}$ to the same resolution. And then, we use the same VAE encoder to obtain the latent of the conditioning image $z_{ref} = \mathcal{E} (I_{ref}) \in \mathbb{R}^{h,w,c}$, where the latent spatial dimensions are downsampled by a factor of 8, i.e., $h = H / 8; w = W/8; \text{ and } c = 4$.
>
> To show how the injection happens during denoising, let’s use a concrete example:
> Let’s say the video frames resolution is: 512 x 512 pixels. Then latent space resolution will be 64x64x4 (after VAE encoding with 8x downsampling). Same operation to the conditional image, which will be preprocessed to a latent that is 64x64x4. The binary mask we have obtained on RGB proxy $\hat{x}_0$ will be downsampled to 64x64 to match latent dimensions. The injection operation happens with:
>
> $x_t^{\text{mix}} = x_t^{\text{video}} \odot (1-m_{64\times64} ) + \lambda \cdot z_{ref} \odot m_{64\times64}.$
>
> When the value in the binary mask is 0, we keep the original content by element-wise multiplication with the original video latent. And when the value in the binary mask is 1, we multiply with the reference image latent, and keep the feature at those 1s region. In the end, we will have a mixed latent of original and conditioned.
>
> When video and reference image have different aspect ratios, we apply center-crop, or bilinear padding to the reference image before encoding to maintain aspect consistency. The VAE’s 8x downsampling ensures that spatial alignment is preserved, which means, a position $(i,j)$ in the mask in pixel space corresponds to the position of $(\lfloor{i/8\rfloor, \lfloor j/8 \rfloor})$ in the latent space.
>
> Regarding the tracking robustness under large appearance changes, we have introduced in the paper relCASS to specifically tackle the evaluation of the large appearance changes performance. Recall that
> $$ \text{relCASS} = \frac{CLIP_{I_{\text{fused}}}-CLIP_{I_{{\text{orig}}}}} {CLIP_{I_{\text{orig}}}} - \frac{CLIP_{T_{\text{fused}}}-CLIP_{T_{{\text{orig}}}}} {CLIP_{T_{\text{orig}}}} $$
>
> A higher relCASS indicates that the fused video significantly increases similarity to the reference image, however it will be mitigated if original image already looks very much like reference image, meaning a big denominator. Similarly, if the original text prompt already a lot like the generated mixed video, then the value will also be mitigated due to a large denominator. The case of large appearance change is represented by small denominator, which means the generated result is nothing like the original, hence it’s small. In our prompt dataset, we have constructed both intra- and inter- class samples, which means in the tested case we already have big appearance change cases, but still in the baseline comparison, we achieve relCASS=1.23, which is 9x better than RAVE, even though it has the second best CASS score, it has low relCASS score, meaning it only works well in similar objects conditioning; and 3x higher than FreeBlend, demonstrating better robustness to large appearance transformation.

---

> ### Author Response · Authors · 2025-11-25
> **Detailed Explanation of FIFO-Diffusion Pipeline and Per-Frame Inference Cost**
>
> To understand how FIFO scheduler is implemented in the video editing process, here we explain explicitly how do we utilize the diagonal denoising scheduler to achieve our goal, which also explains how does it implemented. The diagonal denoising scheduler processes video frames in a queue $Q = \{z_1^{\tau_1}, z_2^{\tau_2}, ..., z_{nf}^{\tau_{nf}}\}$, where each frame $i$ is at a different noise level $\tau_i with 0 < \tau_1 < \tau_2 < ... < \tau_{nf} = T.$ At each iteration, the queue is partitioned into $n$ blocks of size $f$, which is the capacity window size of FIFO diffusion, and each block $Q_k$ is denoisied via $Q_k \leftarrow \Phi (Q_k, \tau_k, c; \theta)$, where $\tau_k = (t_{kf+1}, ..., \tau_{(k+1)f})$. During denoising, we inject reference feature through mask blending $z_i^{mix} = z_i^{video} \odot (1-m_i) + \lambda \cdot z_{ref} \odot m_i.$ After each iteration, the fully denoised frame $z_1^{\tau_0}$ at the queue head is dequeued as the output while a new noisy latent $z_{i+nf}^{\tau_{nf}}$ that initialized with the fully denoised frame $z_1^{\tau_0}$ is enqueued at the tail, creating a sliding temporal window where each frame references both cleaner preceding frames and noisier following frames during denoising. This diagonal pattern enables temporal propagation of semantic edits: as frame i moves through the queue from high noise $\tau_{nf}$ to clean $\tau_0$, it continuously observes already blended neighboring frames, ensuring smooth temporal transition with stable identity alignment across the video sequences. Compared to traditional window-based denoiser, such as Open-Sora, each latent in the window size will be denoised simultaneously at the same step t, instead of at different noise level $\tau_t$, making the temporal relationship vulnerable in identity changed scenario. We have added **Appendix 6.6** to better explain the underlying mechanism of FIFO pipeline
>
> We report per-frame inference time for all compared methods in **Appendix 6.4.1** (excluding preprocessing such as model initialization, base video generation, and feature extraction):
>
> * AnimateDiff: 1s per frame
> * TokenFlow (PnP/SDEdit): 6s per frame
> * FreeBlend + DynamiCrafter: 10s per frame (6s for staged feedback-driven image blending + 4s for video animation)
> * RAVE: 3s per frame
> * AnyV2V: 5s per frame
> * MoCA-Video (ours): 17s per frame, decomposed as:
>     * FIFO-Diffusion diagonal denoising: 13s
>     * MoCA components (IoU tracking + mask extraction from t′ to t=0): 4s
>
> Our method's computational overhead primarily introduced from the diagonal denoising scheduler (13s), which is essential for maintaining temporal coherence during semantic injection. The additional MoCA-specific operations (IoU tracking and segmentation) add only 4s per frame, representing a modest **31%** overhead relative to the base FIFO-Diffusion framework. This overhead is necessary and well-justified: the diagonal denoising design enables forward-referencing across frames, which is critical for achieving smooth semantic mixing with motion preservation, capabilities that faster methods fundamentally lack, as evidenced by their significantly lower CASS scores (see **Table 2**). We have added this to **Appendix 6. 4. 1** for time comparison.

---

### Official Review · Reviewer_WcLF · 2025-11-03

**Soundness:** 2
**Presentation:** 1
**Contribution:** 2
**Rating:** 4
**Confidence:** 4

**Summary:**

This paper proposes a training-free framework (MoCA-Video) for video semantic mixing with latent noise manipulation. IoU-based object tracking in noisy latent, momentum-corrected denoising, and gamma residual stabilization are introduced to achieve the MoCA-Video.

**Strengths:**

Belows are strong points that the paper has:

1. The paper presents thoughtful efforts to enhance video diffusion performance by fusing latents between the original video and the reference image, mitigating motion-induced artifacts through momentum-corrected DDIM, and stabilizing the denoising process with the proposed gamma residual noise.

2. The introduction of an entity blending dataset and the use of task-specific evaluation metrics indicate that the authors carefully designed the experimental setup to assess the proposed model’s performance.

**Weaknesses:**

Belows are weak points that the paper has:

**1. Clarity and Explanation**
-  The paper is difficult to follow due to insufficient self-contained explanations. Specifically, MoCA-Video is built on the denoising scheduler of FIFO-Diffusion, but the paper does not adequately describe FIFO-Diffusion itself. This lack of context makes it challenging for readers unfamiliar with FIFO-Diffusion to fully grasp the proposed methodology.
- Figure 2 is also unclear. The images following DDIM inversion are confusing. it's not evident whether they represent predicted clean $x_{0}$​ images or noisy latents. Moreover, the “Extract Mask Region” step lacks visual or textual clarity, and the meaning of the angle in the “Motion Correction” part is not explained. Without additional clarification, these visual elements are likely to confuse readers.

**2. Limited Model Compatibility**
- The proposed approach is only evaluated with VideoCrafter2, which restricts the generality of the results. In contrast, FIFO-Diffusion validated its framework with multiple video diffusion models. It would strengthen the paper to demonstrate MoCA-Video’s compatibility and performance across other diffusion backbones.

**3. High Sensitivity to Hyperparameters**
- The method relies on several hyperparameters ($\gamma$, IoU threshold, $\beta$, base correction weight $k_{o}$), which may complicate reproduction and tuning. The absence of guidance or ablation on their selection further limits the practical applicability of the approach.

**4.  Lack of Failure Case Analysis**
- The paper does not sufficiently discuss failure cases. For instance, the proposed approach appears to be highly sensitive to the quality of the mask generated by the segmentation model. I also find the authors’ interpretation of Table 4 unconvincing. This table, in fact, suggests that the semantic mixing performance is strongly dependent on mask quality, as evidenced by the substantial performance gap between the CASS and relCASS metrics. A more detailed clarification and analysis of this observation would significantly improve the readers’ understanding of the method’s limitations and robustness.

**Questions:**

Please answer the questions listed in the above Weaknesses section.

---

> ### Author Response · Authors · 2025-11-25
> **Methodology Details Clarification**
>
> We thank the reviewer for the feedback, and raise relevant concerns. We would like to provide further clarification to better alleviate reviewer’s confusion.
>
> The main contribution we adopted from FIFO-Diffusion is its diagonal denoising scheduler,  here we explain explicitly how do we utilize the diagonal denoising scheduler to achieve our goal, which also explains how does it implemented. The diagonal denoising scheduler processes video frames in a queue $Q ={ z_1^{\tau_1}, z_2^{\tau_2}, ..., z_{nf}^{\tau_{nf}} }$, where each frame $i$ is at a different noise level $\tau_i \text{ with } 0 < \tau_1 < \tau_2 < ... < \tau_{nf} = T.$ At each iteration, the queue is partitioned into n blocks of size $f$, which is the capacity window size of FIFO diffusion, and each block $Q_k$ is denoisied via $Q_k \leftarrow \Phi (Q_k, \tau_k, c; \theta)$, where $\tau_k = (t_{kf+1}, ..., \tau_{(k+1)f})$. During denoising, we inject reference feature through mask blending $z_i^{mix} = z_i^{video} \odot (1-m_i) + \lambda \cdot z_{ref} \odot m_i$. After each iteration, the fully denoised frame $z_1^{\tau_0}$ at the queue head is dequeued as the output while a new noisy latent $z_{i+nf}^{\tau_{nf}}$ that initialized with the fully denoised frame $z_1^{\tau_0}$ is enqueued at the tail, creating a sliding temporal window where each frame references both cleaner preceding frames and noisier following frames during denoising. This diagonal pattern enables temporal propagation of semantic edits: as frame $i$ moves through the queue from high noise $\tau_{nf}$ to clean $\tau_0$, it continuously observes already blended neighboring frames, ensuring smooth temporal transition with stable identity alignment across the video sequences. Compared to traditional window-based denoiser, such as Open-Sora, each latent in the window size will be denoised simultaneously at the same step $t$, instead of at different noise level $\tau_t$, making the temporal relationship vulnerable in identity changed scenario. We have added the underlying FIFO mechanism explanation in Appendix 6.6 for better understanding.
>
> Regarding Figure 2, we would like to emphasize that all technical details are currently presented in Section 3.1 - 3.2 and in Alg. 1 and 2. To address the specific concerns that reviewer finds unclear to follow, (1) The output following the DDIM Inversion is correctly the noisy latent $x_t$, consistent with standard diffusion model practices, as detailed in Section 3.1. (2) The "Extract Mask Region" step is a result of the IoU-based object tracking (Algorithm 1) applied to the decoded proxy clean image $x_0$ obtained from the latent, which is thoroughly described in Section 3.1. (3) The angle shown in the "Motion Correction" block represents the direction of the accumulated changing momentum ($v_t$), which is the key mechanism in our momentum-corrected DDIM (Algorithm 2) designed to navigate the denoising trajectory towards novel hybrid distributions, explained in Section 3.2. Beyond the empirical validation of Algorithm 2, we provide theoretical justification in Appendix 6.5 to explain why the heuristic is effective through score function analysis.
>
> Regarding the added hyperparameters, we would like to clarify that the IoU threshold ($\tau$=0.5) and the base correction weight ($\kappa_0$=0.9), momentum decay $\beta=0.9$, gamma residual noise $\gamma=0.05$, are fixed constants throughout our pipeline, we mention it explicitly is to make it clear what we have added beyond the standard DDIM process., the only hyperparameter requiring case-specific tuning is $\lambda$ (blending strength), which governs the intensity of the feature injection from the reference image. We have demonstrated the application of this parameter in **Table 1** (under "Conditioning Strength"), and to ensure maximum transparency and ease of reproduction, upon acceptance, we commit to releasing the full dataset with the corresponding $\lambda$ value used for each specific entity-blending case.

---

> ### Author Response · Authors · 2025-11-25
> **Mask Sensitivity Clarification and Time Efficiency Comparison**
>
> Regarding the concern for segmentation mask sensitivity, in **Table 4** we presented the bounding box experiment establishes the worst-case baseline, however, in actual pipeline, we do not use bounding box during generation. Crucially, even with deliberately imprecise segmentation, MoCA-Video still outperforms all competing methods except for RAVE which uses precise masks, which demonstrates the latent space manipulation has inherent tolerance to masks imperfection. To further valid this claim, we did an additional experiment with ground-truth masks from DAVIS dataset. To quantify explicitly our masks quality, and the difference between our masks with bounding box, we use (a) the ground truth mask; (b) our mask generated during denoising and (c) the bounding box of generated mask as a lower boundary. The results show that IoU scores of $\frac{260.0}{701.0} \approx 0.37 \text{(ground truth mask vs. predicted mask)}\text{ and } \frac{260.0}{1920.0} \approx 0.14 \text{ (ground truth mask vs. bounding box)}\space \frac{701.0}{1920.0} \approx 0.37 \text{(predicted mask vs. bounding box)}$. These metrics confirm that our predicted masks achieve substantially tighter localization compared to bounding boxes  while remaining comparable precision in region detection. The performance gap between precise masks and bounding boxes (CASS: 4.93 → 2.69, a 45% drop) represents the expected degradation from deliberately coarse spatial localization rather than fragility under realistic segmentation conditions. Critically, evaluation on DAVIS ground-truth masks yields CASS scores 4.97, which is only marginally higher than those obtained with our predicted masks, confirming that automatic segmentation provides adequate spatial guidance for semantic injection during denoising. This validates that mask imperfections do not constitute a performance bottleneck, making MoCA-Video practically deployable in scenarios lacking manual annotations
>
> We will report per-frame inference time for all compared methods (excluding preprocessing such as model initialization, base video generation, and feature extraction):
>
> * AnimateDiff: 1s per frame
> * TokenFlow (PnP/SDEdit): 6s per frame
> * FreeBlend + DynamiCrafter: 10s per frame (6s for staged feedback-driven image blending + 4s for video animation)
> * RAVE: 3s per frame
> * AnyV2V: 5s per frame
> * MoCA-Video (ours): 17s per frame, decomposed as:
>     * FIFO-Diffusion diagonal denoising: 13s
>     * MoCA components (IoU tracking + mask extraction from t′ to t=0): 4s
>
> Our method's computational overhead primarily introduced from the diagonal denoising scheduler (13s), which is essential for maintaining temporal coherence during semantic injection. The additional MoCA-specific operations (IoU tracking and segmentation) add only 4s per frame, representing a modest **31%** overhead relative to the base FIFO-Diffusion framework. This overhead is necessary and well-justified: the diagonal denoising design enables forward-referencing across frames, which is critical for achieving smooth semantic mixing with motion preservation, capabilities that faster methods fundamentally lack, as evidenced by their significantly lower CASS scores (see **Table 2**). We have added this to **Appendix 6. 4. 1** for time comparison.

---

### Official Review · Reviewer_8fGT · 2025-11-05

**Soundness:** 2
**Presentation:** 3
**Contribution:** 2
**Rating:** 6
**Confidence:** 3

**Summary:**

This paper introduces MoCA-Video, a training-free framework for video semantic mixing—blending features from a reference image into a target object within a video while maintaining temporal consistency. The method operates in the latent space of a frozen diffusion model (VideoCrafter2). It localizes the target by applying segmentation (Grounded-SAM2) to the predicted clean latent ($\hat{x}_0$) and stabilizes tracking via IoU maximization. To handle the generation of novel, out-of-distribution (OOD) hybrid concepts, the authors propose a momentum-corrected DDIM scheduler, introduced as a heuristic to adjust the denoising trajectory. The paper also introduces a new evaluation metric (CASS) and a benchmark dataset. Experiments show MoCA-Video outperforms existing video editing baselines in semantic blending quality.

**Strengths:**

*   **S1. Novel Task Definition:** The paper addresses the challenging and under-explored problem of temporally consistent video semantic mixing.
*   **S2. Strong Empirical Performance:** The qualitative results are visually compelling. Quantitatively, MoCA-Video significantly outperforms baselines on the proposed CASS metric (4.93 vs. the next best 3.80), which is supported by a user study (Appendix 6.3).
*   **S3. Useful Evaluation Metrics:** The introduction of CASS/relCASS provides a well-motivated metric for evaluating semantic shifts, addressing limitations in prior approaches (Appendix 6.2).

**Weaknesses:**

*   **W1. Lack of Theoretical Rigor:** The core innovation, the momentum correction (Alg. 2), is presented as a heuristic (L213) without theoretical grounding for how it approximates OOD hybrid distributions. The contribution remains largely empirical rather than analytical.
*   **W2. Algorithmic Ambiguity and Missing Details:** Algorithm 2 contains a critical ambiguity: Line 4 defines $g_{t} \leftarrow x_{t}-x_{t-1}+\lambda dir_{t}$. However, $x_{t-1}$ is computed later in Line 7. This ambiguity must be resolved. Additionally, the explicit schedule for the adaptive injection intensity $\lambda$ (L177) is missing, hindering reproducibility.
*   **W3. Severe Computational Overhead:** The inference time is extremely slow. Appendix 6.4 reports 45 minutes to process a 147-frame video on a V100 GPU. This significantly limits practical utility and scalability.
*   **W4. Sensitivity to Segmentation:** The framework's performance is sensitive to the quality of the external segmentation applied to the noisy $\hat{x}_0$. Table 4 demonstrates a significant performance drop (CASS from 4.93 to 2.69) when using coarse bounding boxes, indicating potential fragility in complex scenarios.

**Questions:**

1.  **[W2] Clarification on Algorithm 2:** In Algorithm 2, Line 4: $g_{t} \leftarrow x_{t}-x_{t-1}+\lambda dir_{t}$. Which value of $x_{t-1}$ is used here, given that the updated $x_{t-1}$ for the next step is calculated in Line 7? Does it refer to $x_{t-1}^{(DDIM)}$ calculated in Line 3? Please clarify the notation and the exact update sequence.
2.  **[W1] Theoretical Justification:** Beyond the empirical ablation, can you provide a deeper analysis or theoretical intuition regarding how the "geometric correction" term $x_t - x_{t-1}$ specifically helps reorient the denoising process toward OOD hybrid distributions?
3.  **[W2] Implementation Details:** Please specify the explicit schedule used for the adaptive feature injection intensity ($\lambda$) mentioned in L177.

---

> ### Author Response · Authors · 2025-11-25
> **Theoretical Grounding and Clarification for Alg.2**
>
> We appreciate the reviewer’s clearly stated concerns and thoughtful suggestions. We would like to answer it respectively.
>
> First, regarding the theoretical rigor of Alg.2, the reviewer argues that the heuristic function that we adopted in the Alg.2 is more empirical rather than theoretical proved. As our paper is mainly demonstrate that structured manipulation of diffusion noise trajectories can achieve controllable and high-quality semantic mixed video, experiments in ablation (Sec 4.4.1 Tab. 3) shows that the heuristic Alg.2 module provides substantial improvements in temporal hybrid identity stability, with 39% reduction in LPIPS-T, validating its practical utility. We agree that theoretical grounding can make it more solid, so we would like to provide the approximation proof in our Supplementary:
>
> Let $p_{base}(x_0)$ denotes the data distribution of the original video and $p_{ref}(x_0)$ the distribution of the reference image features. Our goal is to sample from a hybrid distribution $p_{hybrid}(x_0)$ that blends characteristics from both distribution in a spatially localized manner defined by mask $m$.
> For the standard DDIM dynamics, we can approximate $\mathbb{E} [x_0 | x_t]$ under $p_{base}$. The construction of hybrid entity is realized by masked injection, which creates:
> $$ x_t^{\text{mix}} = x_t \cdot (1-\textbf{m}) + \lambda x_t^{\text{cond}} \cdot \textbf{m} $$
> This shifts the $x_t$ towards manifold where the masked region aligns with $p_{ref}$. Here, in response to the reviewer’s last concern, the adaptive feature injection intensity $\lambda = \frac{t}{1000}$, is a time variant variable that adjust the feature injection intensity given the timesteps. It will be stronger in the earlier timestep due to the missing of the fine detail, while weaker in later timestep to avoid overwriting by conditioned features.
>
> In the trajectory correction, the term $g_t = x_t - x_{t-1} + \lambda \text{dir}t$ capture the deviation between the standard denoising direction from $x_{t-1}$ computed from line 4, which also to answer the reviewer’s second confusion, why do we have repeated next timestep latent prediction, it is because that the current $x_{t-1}$ represents the standard next timestep latent prediction that would be under the distribution of $p_{base}(x_0)$, while the line 7 is the predicted $x_{t-1}$ under the distribution of $p_{hybrid}(x_0)$. The term also captures the perturbed direction in $x_t$ in the masked region toward the distribution of $p_{ref}(x_0)$. Specifically, $g_t \approx \Delta x_t^{hybrid} - \Delta x_t^{base}$.
> Under the reverse diffusion process, the instantaneous velocity is governed by the score:
>
> $$\frac{dx_t}{dt} \propto \nabla _{x_t} \log p(x_t) $$
>
> For a discrete scheduler step with step size
>
> $$ \Delta \alpha = \alpha_{t-1} - \alpha_t > 0 $$
>
> Hence, $$ x_{t-1} - x_t \approx c_t \cdot \nabla {x_t} \log p (x_t) + [\text{other terms when substitute to DDIM paper eq (12)}] = \Delta \alpha \cdot [\text{drift terms involving} \nabla{x_t} \log p (x_t)] $$
>
> Hence, $$ g_t \approx \Delta \alpha [\nabla_{x_t} \log p _{hybrid}(x_t) - \nabla _{x_t} \log p _{base}(x_t)] $$
> , which represents the instantaneous score drift at timestep $t$.
>
> By maintaining the accumulative $v_t = \beta v_{t-1} + (1-\beta) g_t$, we compute an exponentially estimate of the persistent directional bias that introduced by the semantic injection, expanding recursively, we have
> $$ v_t = (1-\beta) \sum_{k=0}^{T-t} \beta^{k}g_{t-k} $$ representing an exponentially weighted average of past gradients $\{g_t, g_{t-1}, ..., g_0\}$ During high momentum, when the injection begins, this can term can approximates a temporal smoothing over the trajectory $$ v_t \approx \frac{1}{T-t+1}\sum_{\tau=0}^{t}g_\tau \approx \mathbb{E}{\tau \leq t}[g\tau] $$ Substituting $g_t$ into $v_t$, we have
> $$ v_t \approx \mathbb{E}{\tau \leq t}[\nabla{x_\tau} \log p_{\text{hybrid}}(x_\tau) - \nabla_{x_\tau} \log p_{\text{base}}(x_\tau)] $$, which is the estimated score function drift accumulated along the trajectory.
>
> In the end, the updated term $\hat{x_0}^{(\text{corr})} = \hat{x_0}^{(\text{DDIM})} + \kappa_t v_t$ adjusts the clean image prediction toward the mode of $p_{hybrid}$, where $\kappa_t = \frac{t}{1000}$ providing time dependent weighting that emphasizes corrections in the early denoising, and diminishes them later.
>
> In conclusion, Alg.2 can be proved as a first-order approximation to score matching under distribution shift, where $v_t$ effectively estimates the cumulative score deviation caused by feature injection. While not an exact hybrid sampler, which needs training data for, Alg.2 provides a principled heuristic way that interpolates between known distributions.
>
> We have added Appendix6.5 to include this theoretical grounding perspective, making explicit the connection between momentum correction and score-based approximation of hybrid distribution.

---

> ### Author Response · Authors · 2025-11-25
> **Mask Sensitivity Clarification and Time Efficiency Comparison**
>
> Regarding the concern for segmentation mask sensitivity, **Table 4** where we have the bounding box experiment establishes the worst-case baseline, however, it’s not our intended usage. Crucially, even with deliberately imprecise segmentation, MoCA-Video still outperforms all competing methods except for RAVE which uses precise masks, which demonstrates the latent space manipulation has inherent tolerance to masks imperfection. To further valid this claim, we did an additional experiment with ground-truth masks from DAVIS dataset. To quantify explicitly our masks quality, and the difference between our masks with bounding box, we use (a) the ground truth mask; (b) our mask generated during denoising and (c) the bounding box of generated mask as a lower boundary. The results show that IoU scores of
> $\frac{260.0}{701.0} \approx 0.37 \text{(ground truth mask vs. predicted mask)}\text{ and } \frac{260.0}{1920.0} \approx 0.14 \text{ (ground truth mask vs. bounding box)}\space \frac{701.0}{1920.0} \approx 0.37 \text{(predicted mask vs. bounding box)}$
>
> These metrics confirm that our predicted masks achieve substantially tighter localization compared to bounding boxes while remaining comparable precision in region detection. The performance gap between precise masks and bounding boxes (CASS: 4.93 → 2.69, a 45% drop) represents the expected degradation from deliberately coarse spatial localization rather than fragility under realistic segmentation conditions. Critically, evaluation on DAVIS ground-truth masks yields CASS scores 4.97, which is only marginally higher than those obtained with our predicted masks, confirming that automatic segmentation provides adequate spatial guidance for semantic injection during denoising. This near-equivalence validates that mask imperfections do not constitute a performance bottleneck, making MoCA-Video practically deployable in scenarios lacking manual mask annotations.
>
> Beyond this, the reviewer also poses the concern for complex scenarios performance, we tested on multi-objects scenarios, and report the scores in **Table 5**, which makes segmentation harder due to the multi- identifiable objects. In this experiment, we achieve CASS = 3.74, with 25% degradation still achieves competitive semantic mixing score, showcasing the functionality in complex settings.
>
> We also report per-frame inference time for all compared methods in our in **Appendix 6.4.1**(excluding preprocessing such as model initialization, base video generation, and feature extraction):
> * AnimateDiff: 1s per frame
> * TokenFlow (PnP/SDEdit): 6s per frame
> * FreeBlend + DynamiCrafter: 10s per frame (6s for staged feedback-driven image blending + 4s for video animation)
> * RAVE: 3s per frame
> * AnyV2V: 5s per frame
> * MoCA-Video (ours): 17s per frame, decomposed as:
>     * FIFO-Diffusion diagonal denoising: 13s
>     * MoCA components (IoU tracking + mask extraction from t′ to t=0): 4s
>
> Our method's computational overhead primarily introduced from the diagonal denoising scheduler (13s), which is essential for maintaining temporal coherence during semantic injection. The additional MoCA-specific operations (IoU tracking and segmentation) add only 4s per frame, representing a modest 31% overhead relative to the base FIFO-Diffusion framework. This overhead is necessary and well-justified: the diagonal denoising design enables forward-referencing across frames, which is critical for achieving smooth semantic mixing with motion preservation, capabilities that faster methods fundamentally lack, as evidenced by their significantly lower CASS scores (see **Table 2**).

---

### Comment · Area_Chair_JMQm · 2025-11-27
**Please check the rebuttal**

Dear Reviewers,

The authors have posted their rebuttal. Could you please take a moment to review their responses and check whether your concerns have been adequately addressed if you have done it yet? If possible, kindly initiate the discussion at your earliest convenience.

Your timely assistance is essential for keeping the review process on track. Thank you very much for your support and contribution.

Best regards,
Your AC

---

### Author Response · Authors · 2025-12-02
**Meta-Review Response**

We are deeply grateful to the ACs and reviewers for their generous time, thoughtful feedback, and constructive suggestions throughout the review process. Your feedback substantially improved clarity, technical rigor, reproducibility details, and evaluation completeness.

We are encouraged by positive feedback on novelty, method design, evaluation metrics, and empirical results, such as:  (1) **Novelty**---"novel task definition" (**8fGT**), "training-free approach... controllable hybrid concept mixing" (**hc8D**); (2) **Strong Results**---"visually compelling" (**8fGT**), "CASS/relCASS provides well-motivated metric" (**8fGT**), "wide ablations" (**hc8D**); (3) **Design**---"fusing latents... mitigating motion artifacts through momentum-corrected DDIM" (**WcLF**), "task-specific evaluation metrics" (**WcLF**), "conceptually straightforward combination" (**hc8D**).

We made every effort to address questions, concerns, and comments raised by all reviewers in the rebuttal and discussion. We first address common concerns shared across multiple reviewers, followed by reviewer-specific points.

**Common Concerns Across Multiple Reviewers**
1. Computational Overhead and Runtime Comparison (Reviewers **8fGT, WcLF, hc8D**)

We added per-frame runtime comparison in **Appendix 6.4.1** (excluding preprocessing): AnimateDiff (1s), RAVE (3s), AnyV2V (5s), TokenFlow (6s), FreeBlend+DynamiCrafter (10s), MoCA-Video (17s = 13s diagonal denoising + 4s MoCA operations). Our 31\% overhead over FIFO-Diffusion enables superior temporal coherence and semantic mixing that faster methods lack.

2. Segmentation Mask Sensitivity and Failure Cases (Reviewers **8fGT, WcLF**)

We validated mask robustness using DAVIS ground-truth masks: our predicted masks achieve IoU=0.37 vs. ground-truth (compared to bounding box IoU=0.14), yielding nearly identical performance (CASS=4.93 vs. 4.97 with ground-truth). Even with imprecise bounding boxes (Table 4), MoCA-Video outperforms all methods except RAVE, demonstrating strong tolerance to mask imperfection. Multi-object scenarios (Table 5) show only 25% degradation (CASS=3.74), confirming our automatic segmentation provides adequate guidance without manual annotations.

3. FIFO-Diffusion Background and Implementation Details (Reviewers **WcLF, hc8D**)

We added comprehensive FIFO-Diffusion mechanism explanation in **Appendix 6.6**, detailing the diagonal scheduler's queue-based denoising process with staggered noise levels, reference feature injection through mask blending, and the forward-referencing pattern that enables smooth temporal propagation of semantic edits.

**Response to Reviewer 8fGT's Specific Concerns**: We provide a theoretical justification in **Appendix 6.5**, showing that **Algorithm 2** approximates first-order score matching under distribution shift by introducing a momentum term with exponentially weighted corrections and updated predictions. Regarding algorithmic ambiguity: Line 4 predicts $x_{t-1}$ under base distribution before injection; Line 7 predicts $x_{t-1}$ after injection, computing deviation toward reference features. The adaptive schedule $\lambda = t/1000$ balances strong early injection (high noise) with weaker late injection (avoiding overwriting); missing details are now included.

**Response to Reviewer WcLF:** We address all specific concerns with detailed clarifications. In **Figure 2**, we clarified each component and its corresponded methodology section **(3.1--3.2)**. We also clarified that $\tau=0.5$, $\kappa=0$--$0.9$, $\beta=0.9$, $\gamma=0.05$ are fixed constants (not tunable); only $\lambda$ (blending strength) requires case-specific tuning, demonstrated in **Table 1**. We commit to releasing the full dataset with corresponding $\lambda$ values for reproducibility.

**Response to Reviewer hc8D:** We further clarify implementation details as requested. We emphasize that video semantic mixing fundamentally differs from editing by requiring dual-feature preservation (blending two identities while maintaining motion), a capability lacking in both trained and training-free baselines, justifying our task-specific CASS/relCASS metrics. Key clarifications: (1) we conducted ablation studies on temporal components showing FreeBlend/MagicMix's image-domain limitations; (2) we added comprehensive AnyV2V baseline comparison as suggested, demonstrating frame-by-frame editing cannot achieve dual-feature preservation. We provide detailed reference image alignment workflow in **Appendix 6.1**. All quantitative and qualitative results are updated in **Table 2**, **Table 3**, and **Figure 4**, confirming video semantic mixing as a novel task requiring specialized temporal coherence solutions.

We would like to once again express our sincere thanks to the AC for organizing the review process and to the reviewers for their constructive suggestions. We will incorporate these valuable suggestions into the final version to further improve our work.

---

### Meta-Review · Area_Chair_Vsf8 · 2025-12-29

**Summary:**

This paper presents MoCA-Video, a training-free framework for video semantic mixing that utilizes momentum-corrected diagonal denoising to blend concepts from a reference image into a target video. It solves the problem of maintaining temporal consistency and motion fidelity when injecting novel semantic concepts into dynamic video sequences without model fine-tuning.
The reviewers commend the novel task definition of video semantic mixing, the visually compelling results, and the introduction of well-motivated task-specific evaluation metrics (CASS/relCASS).

The reviewers identified several critical concerns:
1. **Computational Overhead**: Multiple reviewers raised concerns regarding the inference speed (initially cited as 45 minutes for a sequence) and the computational cost compared to other training-free baselines (8fGT, WcLF, hc8D).
2. **Robustness to Segmentation**: Reviewers noted significant performance drops when using bounding boxes instead of precise masks, questioning the method's reliance on high-quality segmentation (8fGT, WcLF).
3. **Clarity and Theoretical Rigor**: Reviewers pointed out a lack of theoretical justification for the Algorithm 2 and insufficient explanation of the underlying FIFO-Diffusion mechanism (8fGT, WcLF).
4. **Novelty**: One reviewer questioned the incremental nature of the contribution relative to existing image mixing (MagicMix, FreeBlend) or video editing (TokenFlow,  AnyV2V) methods (hc8D).

In summary, this paper was reviewed by three experts in the field. The recommendations are 6, 4, 4. The reviewers raised concerns regarding the high computational cost, methodology clarity, and the incremental novelty. After rebuttal, while the paper has merit in its proposed task, concerns on the method's robustness to segmentation quality and the limited technical novelty still remain.

**Reviewer Concerns:**

**Well addressed:**
1. Computational Overhead: The authors provided a detailed breakdown showing a per-frame inference time of 17 seconds. They demonstrated that the 31% overhead compared to FIFO-Diffusion is necessary for the superior temporal coherence achieved. (8fGT, WcLF, hc8D)
2. Theoretical Justification: The authors provided a formal derivation in the appendix showing that their Algorithm 2 approximates first-order score matching under distribution shift. (8fGT)
3. Methodology Clarity: The authors added a comprehensive explanation of the FIFO-Diffusion diagonal scheduler mechanism and resolved ambiguities regarding the update sequence in Algorithm 2. They also clarified hyperparameter settings. (WcLF, 8fGT)
4. Baseline Comparisons: The authors added a comparison with AnyV2V as requested, demonstrating that frame-by-frame editing followed by propagation fails to achieve the dual-feature preservation that MoCA-Video offers. (hc8D)

**Partly addressed:**
1. Sensitivity to Segmentation: The authors conducted additional experiments using DAVIS ground-truth masks to show that their generated masks achieve comparable performance to ground truth. While the performance drop with bounding boxes remains, the authors argued that their method still outperforms baselines. This mitigates the concern, though reliance on segmentation remains a characteristic of the pipeline. (8fGT, WcLF)
2. Novelty: The authors argued that video semantic mixing is a distinct task from standard editing, requiring the preservation of both the original motion and the new reference identity. While they validated this with metrics, the perception of incremental contribution vs. novel task is subjective and may arguably remain for the reviewer. (hc8D)

**Unsolved:**

None

**Reviewer Scores:**

**8fGT (6):**

This reviewer was already positive but concerned about theoretical justification and computational overhead. The authors provided the requested theoretical proof for Algorithm 2 and the clarification on inference time. Given that the Weaknesses were directly addressed, the reviewer would likely maintain or raise their score to 8.

**WcLF (4):**

This reviewer's main issues were the mask sensitivity. Since the fundamental concern about the method's reliance on high-quality segmentation remains, the score might remain at 4.

**hc8D (4):**

This reviewer focused on incremental novelty and efficiency. The authors provided a runtime comparison which justifies the cost against the quality gain and effectively differentiated their task from standard video editing. However, the reviewer's concern about the incremental nature of the contribution is strong. The score likely remains at 4.

---

### Decision · Program_Chairs · 2026-01-26

Reject